# The Immunomodulatory Effects of Fluorescein-Mediated Sonodynamic Treatment Lead to Systemic and Intratumoral Depletion of Myeloid-Derived Suppressor Cells in a Preclinical Malignant Glioma Model

**DOI:** 10.3390/cancers16040792

**Published:** 2024-02-15

**Authors:** Serena Pellegatta, Nicoletta Corradino, Manuela Zingarelli, Edoardo Porto, Matteo Gionso, Arianna Berlendis, Gianni Durando, Martina Maffezzini, Silvia Musio, Domenico Aquino, Francesco DiMeco, Francesco Prada

**Affiliations:** 1Unit of Immunotherapy of Brain Tumors, Fondazione IRCCS Istituto Neurologico Carlo Besta, Via Celoria, 11, 20133 Milan, Italy; manuela.zingarell@polimi.it (M.Z.); arianna.berlendis@istituto-besta.it (A.B.); martina.maffezzini@istituto-besta.it (M.M.);; 2Unit of Neuro-Oncology, Fondazione IRCCS Istituto Neurologico Carlo Besta, 20133 Milan, Italy; 3Department of Neurological Surgery, Fondazione IRCCS Istituto Neurologico “C. Besta”, Via Celoria 11, 20133 Milan, Italy; nicoletta.corradino@unimi.it (N.C.); edoardo.porto@gmail.com (E.P.); francesco.dimeco@istituto-besta.it (F.D.); 4Department of Oncology and Hemato-Oncology, University of Milan, 20122 Milan, Italy; 5Acoustic Neuroimaging and Therapy Laboratory (ANTY-Lab), Fondazione IRCCS Istituto Neurologico Carlo Besta, 20133 Milan, Italy; matteo.gionso@gmail.com (M.G.); g.durando@inrim.it (G.D.); 6Focused Ultrasound Foundation, Charlottesville, VA 22903, USA; 7Department of Neurosurgery, School of Medicine, Emory University, Atlanta, GA 30322, USA; 8Department of Biomedical Sciences, Humanitas University, Via Rita Levi Montalcini 4, Pieve Emanuele, 20072 Milan, Italy; 9Istituto Nazionale di Ricerca Metrologica, 10135 Turin, Italy; 10Unit of Neuroradiology, Fondazione IRCCS Istituto Neurologico Carlo Besta, 20133 Milan, Italy; domenico.aquino@istituto-besta.it; 11Department of Neurological Surgery, Johns Hopkins Medical School, Hunterian BrainTumor Research Laboratory CRB2 2M41, Baltimore, MD 21231, USA; 12Department of Neurological Surgery, University of Virginia Health System, Charlottesville, VA 22903, USA

**Keywords:** sonodynamic therapy, fluorescein, ultrasound, glioma, immune microenvironment, mouse model, MDSC, CD8 infiltrating T cells

## Abstract

**Simple Summary:**

Sonodynamic therapy (SDT) is emerging as a promising innovative technique for treating malignant gliomas in a noninvasive fashion. The use of fluorescein, instead of 5-aminolevulinic acid, could extend the application of sonodynamic therapy to intracranial tumors other than gliomas. Our aim was to investigate the feasibility of sonodynamic therapy with fluorescein in an intracranial mouse model of malignant glioma, its efficacy, and its effects on the immune microenvironment.

**Abstract:**

Fluorescein-mediated sonodynamic therapy (FL-SDT) is an extremely promising approach for glioma treatment, resulting from the combination of low-intensity focused ultrasound (FUS) with a sonosensitizer. In the present study, we evaluated the efficacy and immunomodulation of SDT with fluorescein as the sonosensitizer in immunocompetent GL261 glioma mice for the first time. In vitro studies demonstrated that the exposure of GL261 cells to FL-SDT induced immunogenic cell death and relevant upregulation of MHC class I, CD80 and CD86 expression. In vivo studies were then performed to treat GL261 glioma-bearing mice with FL-SDT, fluorescein alone, or FUS alone. Perturbation of the glioma-associated macrophage subset within the immune microenvironment was induced by all the treatments. Notably, a relevant depletion of myeloid-derived suppressor cells (MDSCs) and concomitant robust infiltration of CD8+ T cells were observed in the SDT-FL-treated mice, resulting in a significant radiological delay in glioma progression and a consequent improvement in survival. Tumor control and improved survival were also observed in mice treated with FL alone (median survival 41.5 days, *p* > 0.0001 compared to untreated mice), reflecting considerable modulation of the immune microenvironment. Interestingly, a high circulating lymphocyte-to-monocyte ratio and a very low proportion of MDSCs were predictive of better survival in FL- and FL-SDT-treated mice than in untreated and FUS-treated mice, in which elevated monocyte and MDSC frequencies correlated with worse survival. The immunostimulatory potential of FL-SDT treatment and the profound modulation of most immunosuppressive components within the microenvironment encouraged the exploration of the combination of FL-SDT with immunotherapeutic strategies.

## 1. Introduction

Glioblastomas are still the most urgent unmet needs in oncology. The immune landscape of glioblastoma is characterized by a myeloid-enriched and immunosuppressive microenvironment, which is demonstrated by their resistance to therapies, including immunotherapy [1,2,3,4,5,6].

Combinatorial strategies designed to modulate the immunosuppressive microenvironment and increase tumor immunogenicity are urgently needed. Recently, as a noninvasive treatment modality, sonodynamic therapy (SDT) has gradually attracted increased amounts of attention. Several recent studies have reported that SDT can induce apoptosis and necrosis [7,8,9] but also favor immunogenic cell death, a process during which tumor cells release specific warning signals (i.e., calreticulin, ATP, and HMGB1), which are implicated in immune cell activation and potential long-term antitumor immunity [10,11].

Sonodynamic therapy (SDT), an application of focused ultrasound (FUS), is an extremely promising technique for cancer treatment based on the combination of a sonosensitizer compound and targeted low-intensity FUS to generate reactive oxygen species (ROS) able to damage tumor cell DNA, thus inducing apoptosis [9,12,13]. For this reason, the efficiency of SDT is determined by the performance of the sonosensitizer.

The sonosensitizers employed are nontoxic chemical compounds whose therapeutic activity is triggered by ultrasound. Ideal sonosensitizers should have high selectivity for the target lesion, a high clearance rate from healthy tissue, and no substantial in vivo toxicity [14]. To date, 5-aminolevulinic acid (5-ALA) and fluorescein are the two main investigated compounds for the application of SDT in neuro-oncology. These compounds accumulate selectively in brain tumors and show good safety. Both are already approved for human use in image-guided surgery for high-grade gliomas, hence making SDT a realistic advanced approach [15,16,17,18]. The main limitation of 5-ALA is the field of application: tumors other than high-grade gliomas (HGG) do not exhibit uptake.

Fluorescein (2-(6-hydroxy-3-oxo-(3H)-xanthen-9-yl) benzoic acid) (FL) is a biosafe organic xanthene-based dye that is consolidated in neuro-oncology for the removal of HGG and is also routinely used for the resection of metastatic brain tumors [15]. FL preferentially accumulates in brain areas where the blood-brain barrier (BBB) is disrupted, while it rapidly washes out from vessels and the healthy brain; this makes FL a suitable compound for image-guided surgery during glioma resection [19,20,21]. A multicenter phase II study demonstrated that fluorescein-guided resection of HGGs is safe and allows complete resection via early postoperative MRI [22].

SDT with FL was tested for the first time in vivo in a rat-C6 subcutaneous glioma model, in which a soluble and time-dependent accumulation of tumor mass was observed, with a peak concentration occurring between 15 and 30 min after administration. Furthermore, compared with the control and FUS-only, FL-SDT significantly inhibited tumor growth [23].

Notably, a recent study demonstrated that peripheral blood leukocytes, macrophages, and glioma cell lines can take up FL and that the accumulation of FL can increase over time. It was also observed, via analysis of different types of human tumor specimens and glioblastoma cells, that FL accumulates intracellularly in different immune cell subsets within the tumor microenvironment (TME). The highest accumulation of these cells was observed in myeloid cells, which include bone marrow-derived macrophages and microglia [24].

All these observations represent a solid prerequisite for investigating the crucial contribution of FL-SDT to activating antitumor immunological effects.

In the present study, we explored the use of SDT in combination with FL as a new strategy for combating immune suppression in the microenvironment, with particular attention given to the myeloid compartment, and explored its immunostimulatory potential in immunocompetent malignant glioma-bearing mice [25,26].

## 2. Materials and Methods

### 2.1. Cell Culture

GL261 cells were cultured as neurospheres in DMEM-F12 Glutamax, B-27 supplement (ThermoFisher Scientific, Waltham, MA, USA), penicillin/streptomycin (ThermoFisher Scientific, Waltham, MA, USA, human recombinant epidermal growth factor (EGF; 20 ng/mL, ThermoFisher Scientific, Waltham, MA, USA, and human recombinant fibroblast growth factor-2 (ThermoFisher Scientific, Waltham, MA, USA).

### 2.2. Apoptosis Assay

GL261-NS were treated with FL either alone or combined with SDT. Cells were seeded at 3 × 10^5^ cells/well and collected 12, 24, and 48 h after treatment for apoptosis measurement. Early and later apoptosis were distinguished with Annexin V positivity and Annexin V-PI double positivity, respectively. Apoptosis and necrosis were measured using the CellEvent™ Caspase 3/7 Green Flow Cytometry Assay Kit (Life Technologies, Carlsbad, CA, USA).

### 2.3. RNA Extraction and Real-Time PCR

Total RNA was extracted using Trizol reagent (Life Technologies, Carlsbad, CA, USA) from GL261-NS collected at 12, 24, and 48 h after FUS treatment. Total RNA was reverse-transcribed using a high-capacity cDNA synthesis kit (ThermoFisher Scientific, Waltham, MA, USA). The expression of Rae-1 isoforms MULT-1, MICA, and MICB were detected by SybrGreen chemistry performed on a ViiA7 Real-Time PCR system (Life Technologies, Carlsbad, CA, USA) and normalized relative to beta-actin. The RNA of FL-GL261-NS was used as the calibrator for the calculation of fold expression levels with the ΔΔCt method.

Rae1b fw CAGCAAATGCCACTGAAGTGAA rev GGTCTTGTGAGTGTCCACTTTGRae1d fw CTCCTACCCCAGCAGATGAAG rev CCCTGGGTCACCTGAAGTCRae1e fw GACCCACAGACCAAATGGCA rev CTCTGTCCTTTGAGCTTCTTGCMICA fw CCACCTGTGGATAGTGTACCTG rev GCCACCAGTCTTTGGTTGTCMICB fw GTTTCTGGCTGACGTGGAG rev ATAGCGCAGAGTGTGGGTTCMULT fw TGAAGTCACCTGTGTTTATGCAG rev CACTGTCAAAGAGTCATCCAACAβ-actin fw GATGTGGATCAGCAAGCAGGA rev AGCTCAGTAACAGTCCGCCTA

### 2.4. Mice

C57BL/6N 6-week-old female mice (Charles River Laboratories) received intracranial injections of 0.5 × 10^5^ GL261-NS using specific stereotactic coordinates into the nucleus caudatum (0.7 mm posterior, 3 mm left lateral, and 3.5 mm deep, with respect to the bregma). Mice were divided into four groups and treated 9 days after GL261 implantation with FUS-only, FL-only, FL-SDT, and no treatment. A total of 6 mice/group were monitored by MRI at three different time points (14, 21, and 28 days after tumor implantation), and followed for survival analysis. Blood samples were collected and analyzed at the same time points. Separated subgroups were studied to evaluate the immune infiltration into the tumor. The animals were monitored every day and euthanized when suffering, in accordance with the current directives of the Fondazione IRCCS Istituto Neurologico Carlo Besta house facility and the Minister of Health. Animal experiments were performed in accordance with the Italian Principle of Laboratory Animal Care (Lgs. 26/2014) and European Communities Council Directives (86/609/EEC and 2010/63/UE).

### 2.5. Sonodynamic and Other Treatments

In vitro experiment—After seeding a total of 0.3 × 10^6^ GL261 in 35 mm^2^-plates and adding FL (0.5 mg/mL), FUS was applied continuously for 20 min at a frequency of 0.983 MHz, 100 kPa of peak positive pressure, 10% duty cycle (300 cycles, 10 ms period). Then cells were incubated for 12, 24, and 48 h, and then analyzed for immunogenicity and apoptosis.In vivo experiment—In the FL-SDT group, after anesthesia, sodium fluorescein 10 mg/kg was injected intraperitoneally, and mice were sonicated 20 min later. To perform low-intensity-focused ultrasound, we used a single-element planar transducer with 0.485 Freq./MHz, 100 kPa of peak positive pressure, 10% duty cycle (350 cycles, 10 ms period) for 20 min. These parameters were maintained for the FUS-only group and the same FL dose was used in the FL-only group. The SDT was performed using a plane wave source transducer TRA08 (Istituto Nazionale di Ricerca Metrologica, Turin, Italy) based on a lithium-niobate piezoelectric transducer (Boston Piezo-Optics Inc., Bellingham, MA, USA), connected to a signal generator (Model 33250A, Agilent Technologies, CA, USA) through a power amplifier (Model AR 100A250A, Amplifier Research, PA, USA). The animal was anesthetized using tribromoethanol and positioned prone on a soft pad. The transducer was positioned close to the mouse’s skull, at the level of the tumor injection point. A US aqueous coupling medium was used to reduce air interference to a minimum. Images of the setup are reported in the Appendix A.

### 2.6. MRI

In vivo MRI was performed by a horizontal-bore preclinical scanner (BioSpec 70/20 USR, Bruker, Ettlingen, Germany). The system had a magnetic field strength of 7 T (1H frequency 300 MHz) and a 20 cm bore diameter. All acquisitions were carried out using a cross coil configuration: a 72 mm linear birdcage coil was used for radiofrequency excitation and a mouse brain surface coil received the signal.

Mice were anaesthetized with 1.5–2% isoflurane (60:40 N_2_O:O_2_ (vol:vol), flow rate 0.8 L/min). The depth of anesthesia and the animal health condition were detected during the acquisition by monitoring the respiratory rate using a pneumatic sensor.

GL261 glioma-bearing mice treated and untreated underwent high-resolution MRI investigation at four different time points: day −1 (one day before treatments), 14, 21, and 28. The following protocol was performed: a T2-weighted Rapid Acquisition with Reduced Echoes (RARE) sequence (TR = 3360 ms, TE = 35 ms, in plane resolution = 100 × 100 µ µm^2^, slice thickness = 400 µm, 4 averages, total acquisition time of 5 min 36 s) and T1-weighted RARE sequences (TR = 510 ms, TE = 8 ms, in plane resolution = 78 × 78 µm^2^, slice thickness = 400 µm, 6 averages, total acquisition time of 9 min 47 s) were acquired after intraperitoneal administration of gadolinium-based contrast medium. All sequences were acquired along the same coronal geometry (400 µm thick continuous slices), with slice package posterior to olfactory bulb and anterior to cerebellum.

T2 and post-contrast T1 sequences were selected in order to rule out complications such as cerebral oedema, hemorrhages, ischemia, and hematomas, and to evaluate tumor progression.

### 2.7. Isolation of CNS-Infiltrating Lymphocytes

Infiltrating immune cells were isolated using a tumor dissociation kit (mouse, Miltenyi Biotec) on days 14, 21, and 28 after tumor implantation. Gliomas were explanted from untreated and treated mice (*n* = 3/group), cut into small pieces of 2–4 mm, and dissociated using GentleMACS (Miltenyi Biotec) according to manufacturer’s instruction. The cells were then suspended in PBS/0.5% bovine serum albumin/2 mM EDTA buffer for labeling and flow cytometry evaluation.

### 2.8. Flow-Cytometer Analysis

Peripheral blood leukocytes from peripheral blood and infiltrating immune cells isolated from the explanted gliomas of untreated and treated mice, were used for immune monitoring. Briefly, cells were stained in PBS-BSA for 10 min at RT with the following antibodies: anti-CD3, -CD8, -CD4, -CD45, Gr1, and CD11b.

## 3. Results

### 3.1. In Vitro Effects of FL and FL-SDT Treatment on GL261 Glioma Cells

To evaluate the ability of FL-SDT to induce immunogenic apoptosis, also termed immunogenic cell death (ICD), an in vitro study was performed on murine GL261 glioma cells cultured as neurospheres (GL261-NS). After FUS exposure of GL261-NS cells previously treated with FL, early and late apoptosis were assessed by flow cytometry at three different time points (12, 24, and 48 h) (Figure 1A). A significant increase in both early and late apoptosis was observed in the GL261-NS cells compared to those in the control cells treated with FL only (Figure 1B–D). A more specific analysis based on Caspase 3/Caspase 7 detection was performed to clearly identify the live and dead populations; the results revealed not only apoptotic but also necrotic cells, which were significantly increased at 48 h (Figure 1E,F).

ICD was confirmed by evaluating the flow cytometry expression of MHC class I (Figure 2A,B) and the costimulatory molecules CD80 and CD86 (Figure 2C–F). FUS treatment of FL-pretreated GL261-NS cells induced a significant alteration in the phenotype that was not observed upon individual exposure to FL. In particular, at 12 h, MHCI, CD80, and CD86 expression was unaffected by the treatment, but CD80 and CD86 expression was significantly increased at 24 and 48 h (Figure 2).

The expression of the NKG2D ligands Rae-1 isoforms (Figure 2G–I) and MULT-1 (Figure 2J) were susceptible to FUS treatment, as evaluated by real-time PCR. In particular, the expression of these molecules started to increase significantly after 24 h and was homogeneously upregulated at 48 h. Therefore, MICA and MICB, the stress-inducing molecules recognized by NKG2D and the major activation receptors of NK cells, exhibited the same expression kinetics, supporting the immunomodulatory effect of in vitro FL-SDT treatment (Figure 2K,L).

### 3.2. In Vivo Effects of FL and FL-SDT Treatment

The effects of FL alone and FL-SDT were evaluated in vivo in GL261 glioma-bearing mice (Figure 3A). Mice were treated 9 days after intracranial implantation of GL261-NS cells.

Magnetic resonance imaging (MRI-7 Tesla Bruker) was performed one day before treatment (day 8) to select mice with similar tumor engraftment. Mice were randomized into four different groups: FL only, FUS only, FL-SDT, and untreated. All mice were monitored once a week for 2–3 weeks.

The 7T-MRI performed at 14, 21, and 28 days showed no treatment-related complications, such as hemorrhages or hematomas. T2 and T1 with gadolinium enhancement imaging showed macroscopic delay or inhibition of tumor growth in the FL-SDT group compared to that of the FUS only, FL only (Figure 3B–D), and untreated groups. The endpoint for assessing treatment efficacy was determined by measuring the tumor size via MRI relative to the best response scan. Rapid tumor progression was observed in the untreated mice, which all died within 21 days (median survival 19 days), with a maximum tumor diameter of approximately 6.7 mm. Mice treated with FL-SDT showed the most evident antitumor effect, as indicated by the absence of lesions on day 28, but smaller and more circumscribed masses were also observed in mice treated with FL only (median survival 41.5 days, *p* > 0.0001 compared to untreated mice). Tumor control was also observed with FUS only, even if the antitumor effect was less dramatic, as observed by tumor diameter and survival, since no mice survived longer than 32 days (median survival 29.5 days). 

TME-infiltrating immune cells isolated from untreated and treated mice were analyzed by flow cytometry and identified as CD45+ cells. The glioma-associated macrophages/microglia (GAM) compartment, assessed for the expression of CD11b, included resting (CD45low/CD11b+) and activated (CD45high CD11b+) cells. MDSCs, the most immunosuppressive component within the TME, were identified as CD11b+/Gr-1+ cells within the activated GAM subset. When present, the lymphocytes were CD45high CD11b- (Figure 4A).

Untreated GL261-type gliomas were strongly infiltrated by activated GAMs, also maintaining the resting GAM component (Figure 4B). In FUS-, FL-, and FL-SDT-treated GL261-type gliomas, the resting subset was limited, exhibiting 4.7-, 5.2-, and 12-fold decreases, respectively (*p* < 0.001, compared to untreated), while the activated subset was more prevalent, with a 1.6-fold increase in FL-SDT compared to the untreated group (*p* < 0.05) (Figure 4B). The most relevant difference among the untreated and treated groups was related to the MDSC infiltrate measured within the activated component. MDSCs constituted the dominant population of activated GAMs in untreated mice, exhibiting a very high percentage of MDSCs (>80%) at earlier time points, demonstrating the role of these cells in sustaining tumor progression (Figure 4C). At the same time points, the proportion of MDSCs was significantly lower in the treated mice, particularly in the FL and FL-SDT groups (Figure 4C). A progressive time-dependent significant increase in MDSCs was revealed in FUS and FL patients only (*p* < 0.0001 on day 28 compared with day 14). In the FL-SDT-treated group, the frequency of MDSCs was consistently low, even on day 28 (Figure 4C), supporting the absence of tumor engraftment, as evaluated by MRI.

CD3+ T cells within the CD45+ immune infiltrate were rare in untreated mice and exhibited significant 2.7-, 3.3-, and 11-fold increases in FUS-, FL-, and FL-SDT-treated mice, respectively (Figure 4B).

Considering the ability of MDSCs to specifically inhibit CD8+ T cells, a focused analysis was performed to assess their infiltration. CD8+ TILs were detectable in untreated mice on day 14 and showed a rapid and significant decrease within one week. At the same time, compared with those in untreated gliomas, CD8+ T cells were highly infiltrated in treated gliomas (1.9-, 2, 1-, and 2.5-fold greater in FUS only, FL only, and FL-SDT, respectively) (Figure 4D). At later time points, the CD8+ T-cell infiltration significantly decreased in the FUS-only and FL-only groups (38.2 ± 3.4 to 19.2 ± 5.2 and 44.5 ± 3.0 to 34.5 ± 5.0, respectively) (Figure 4D), while maintaining high and almost unmodified levels in the FL-SDT cohort, suggesting that this treatment is efficient at modulating the MDSC compartment and promoting the infiltration and expansion of CD8+ T cells.

MDSCs and T cells were also examined in the peripheral blood of untreated and treated mice at different time points. Monocytes were discriminated from lymphocytes by flow cytometry based on the expression level of CD45 and the SSC parameters (Figure 4E). Notably, a higher proportion of monocytes was observed in the blood of untreated and FUS-only treated mice than in that of FL-only and FL-SDT-treated mice, in which the most abundant CD45+ cells were CD3+ T cells (Figure 4E,G). The MDSC percentage was impressively high not only in the untreated group but also in the FUS-only treated group (Figure 4E,F), remaining very low in the FL-only and FL-SDT-treated groups, showing a progressive increase at later time points, albeit less evident for the FL-SDT group until day 42 (Figure 4F), when the percentage of T cells underwent a slight but nonsignificant decrease indicating the most prolonged survival. (Figure 4G).

Our data and previous results encouraged us to consider the ability of systemic increases in MDSCs to predict worsening of tumor progression.

## 4. Discussion

Our results showed that sonodynamic treatment with fluorescein is efficient at modulating the tumor microenvironment and improving the survival of glioma-bearing mice. The most significant immunostimulatory effects were observed on the myeloid component within the TME. The myeloid-derived suppressor cells, previously identified as one of the most immunosuppressive subsets exerting specific inhibitory effects on CD8+ T cells, were depleted and maintained at a very low level over time. This depletion resulted in significantly increased CD8+ T-cell infiltration and prolonged survival. When FUS and FL were administered as single treatment modalities, they exhibited some immunomodulatory effects, with a specific impact on MDSC depletion.

In FUS−treated mice, MDSC depletion was significant at earlier time points. However, the effect did not persist, reflecting more rapid tumor progression and, albeit significantly, shorter survival than those of FL−only and FL−SDT-treated mice.

As previously reported, SDT can create hypoxia within the tumor microenvironment because of oxygen depletion. In a model of breast cancer, oxygen consumption during the SDT process was shown to induce exacerbation of hypoxia, resulting in increased recruitment of MDSCs within the tumor and the TME [27]. In the same study, a sono-activatable immunotherapeutic strategy appropriately designed by encapsulating a hydrophobic hypoxia-responsive tirapazamine (TPZ)-conjugate activated for increasing immunogenic cell death and a drug (ibrutinib) for reducing the recruitment of MDSCs [27] was used to efficiently maintain the primary tumor and limit metastasis. Based on these data and the evidence supporting the strong correlation between hypoxia and MDSC recruitment in glioma [28,29], a similar process might also be implicated in our model, which showed a progressive increase in MDSCs and a consequent failure of FUS-only treatment.

The most surprising result was that when FL was administered as a single treatment, the TME was strongly modulated, as indicated by a low percentage of MDSCs and robust infiltration of CD8+ T cells lasting for several weeks; moreover, peripheral leukocytes were also influenced. The lymphocyte–monocyte ratio was altered to favor lymphocytes, and the percentage of MDSCs was significantly lower in the treated mice than in the untreated and FUS−only treated mice. The augmentation of peripheral MDSCs at later time points corresponded with their increased infiltration within the TME, which impacted tumor progression. A similar result was observed in mice treated with FL−SDT, in which the efficacy was higher than that observed in mice treated with FL alone. The intratumoral and peripheral frequencies of MDSCs were decreased for a longer duration, which correlated with massive infiltration of CD8+ T cells, prolonged tumor control, and improved survival.

A study performed by Chae and colleagues on GL261 glioma reached the conclusion that intratumoral and systemic MDSCs can arise from monocytes after they undergo immunosuppressive re-education. Increased intratumoral and systemic MDSC levels are correlated with decreased survival and inevitably with a failure of the antitumor immune response [30]. The impact observed on peripheral immune cells after FL administration can be explained by the ability of leukocytes to take up and accumulate FL, as very recently described by Musca and coworkers [3]. The data from these studies and our study need to be further investigated to clarify the immunostimulatory potential of FL, either alone or in combination with SDT.

Although our results are promising, it is necessary to declare the limitations of our study. The survival of mice treated with FL−SDT was significantly longer than that of untreated mice and mice treated with other combinations. However, despite the evidence of tumor control and late engraftment, as observed by MRI, no mouse survived longer than 68 days. The immunostimulatory effects observed are promising for cancer immunotherapy, including dendritic cell immunotherapy and adoptive cell therapy (CAR−T and TIL therapy).

In our study, we used a preclinical single-element planar transducer (0.485 Freq./MHz) to perform ultrasounds; even if, until recently, low-intensity focused ultrasound was administered to humans through MRI-guided focused ultrasound, 4000 Insightec’s MRgFUS was approved. However, even though bone is a barrier for ultrasound waves, we should not exclude the possibility of SDT becoming a bedside treatment. In this respect, Prada et al. produced an ultrasound-compatible artificial operculum (In. Tra. Prothesis) made from high-density, low-porosity homogenous material, and their biocompatibility was certified. It is designed, patented, and tested for minimum attenuation and distortion, thus allowing both transcranial imaging and therapeutic ultrasound applications. It is worth considering that low-intensity US delivery through this prothesis could contribute to the clinical application of SDT [31].

In our study, SDT treatment was performed without microbubble considering that the energy provided by low-intensity focused ultrasound can activate fluorescein. Chemically, the main mechanism of action is that the energy released leads to the production of reactive oxygen species (ROS), starting a peroxidation chain that leads to necrosis and apoptosis of tumor cells. However, SDT mechanisms of action are still under investigation.

In general, as a noninvasive treatment with the potential to modulate the tumor and immune microenvironment, FL−SDT is also a promising method for treating glioblastoma, a rare but serious and untreatable cancer that is still an unmet medical need in oncology.

## 5. Conclusions

SDT with FL and low-intensity focused ultrasound is a feasible technique for in vivo treatment of deep-seated intracranial high-grade gliomas in orthotopic mouse models. In this regard, this study assessed the effects of sonication parameters and FL doses on FL-SDT in an intracranial brain tumor model established in GL261 glioma-bearing mice.

In our study, FL-SDT was proven to be safe and feasible in this model. Moreover, FL-SDT has been associated with glioma growth inhibition, as well as with impressive modification of the TME, reducing MDSC numbers and enhancing CD8+ infiltration.

The use of FL in SDT instead of other sonosensitizers (5-ALA or sinoporphyrin sodium) could provide advantages in terms of SDT application to different types of brain tumors, such as meningiomas or chordomas, and in terms of glioma microenvironment modulation. Considering the specific effect of FL-SDT in reducing MDSCs and increasing T-cell CD8+ infiltration, its future application in clinical trials could be coupled efficiently with immunotherapy. Because FL is an FDA-approved compound for human use, FL-SDT is expected to be applied clinically for deep-seated tumor treatment in humans in the near future.

## Figures and Tables

**Figure 1 cancers-16-00792-f001:**
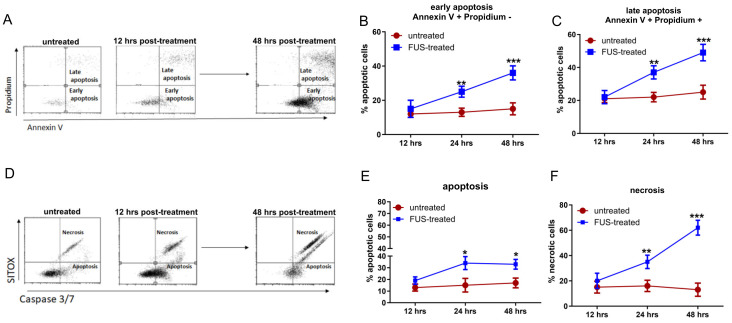
(**A**) Representative flow cytometry dot plots of Annexin V-PI-stained GL261-NS cells. GL261-NS cells incubated with FL only or treated with FL-SDT were analyzed for cell death. Each plot shows cells positive for Annexin V only (LR), positive for PI only (region UL), and negative for both (region LL), representing early apoptotic, late apoptotic, and live cells, respectively. (**B**,**C**) The graphs show the percentage of early (**B**) and late (**C**) apoptotic cells measured after 12, 24, and 48 h (red line, FL-only or “untreated”; blue line, FL-SDT-treated or “FUS-treated”). (**D**) Representative flow cytometry dot plots for Caspase 3/Caspase 7 indicating live cells (LL), late apoptotic cells (LR), and necrotic cells (UR). (**E**,**F**) The graphs show the percentages of apoptotic (**E**) and necrotic (**F**) cells measured after 12, 24, and 48 h (red line, FL-only; blue line, FL-SDT-treated cells). (* *p* < 0.01, ** *p* < 0.005, *** *p* < 0.0001; FL + SDT vs. FL only). The data from three experiments are presented as the mean ± SD.

**Figure 2 cancers-16-00792-f002:**
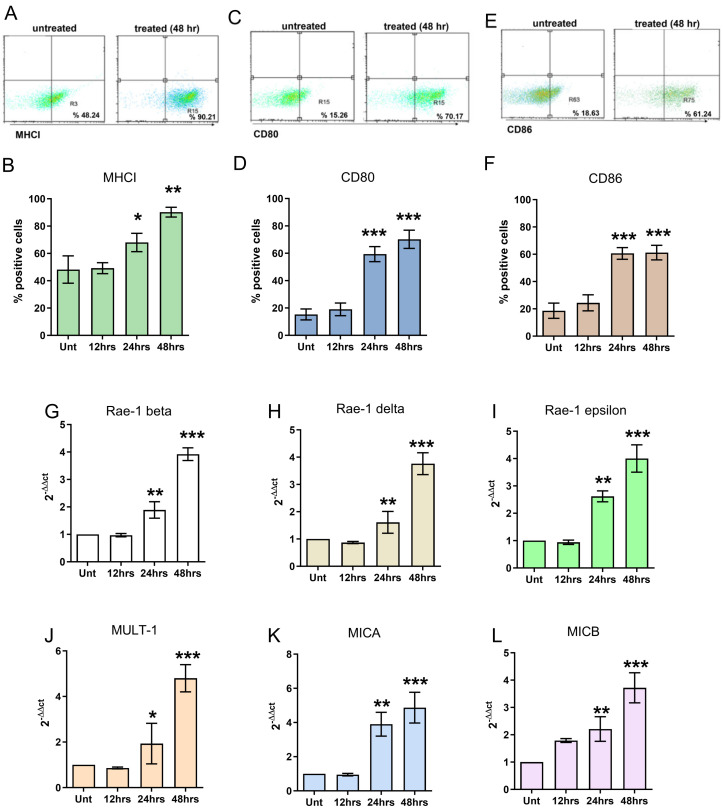
(**A**,**C**,**E**) Representative flow cytometry dot plots showing the expression of MHCI (**A**), costimulatory CD80 (**C**), and CD86 (**E**) on GL261-NS cells treated with FL-SDT compared with that on cells incubated with FL. (**B**,**D**,**F**) Bar graphs showing the percentage of positive cells as assessed by flow cytometry at 12, 24, and 48 h after FUS treatment compared with that of untreated but FL-preincubated GL261-NS cells. (**G**–**L**) Bar graphs showing the relative expression of different NKG2D ligands at different time points after FL-SDT treatment compared to that in untreated cells (* *p* < 0.01, ** *p* < 0.005, *** *p* < 0.0001; FL + SDT vs. FL only). The data from three experiments are presented as the mean ± SD.

**Figure 3 cancers-16-00792-f003:**
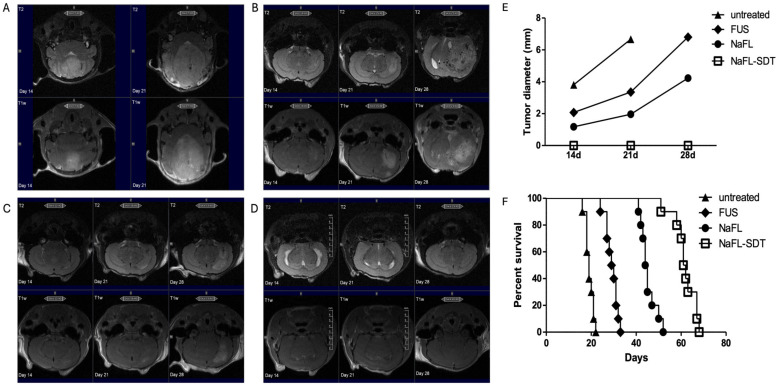
(**A**–**D**) Representative MRI images performed at three different time points with T2-weighted images (T2-wi) and T1-weighted images (T1-wi) with contrast medium injection showing the pattern of tumor progression in untreated control mice (**A**), treated with FUS only (**B**), sodium fluorescein (NaFL) only (**C**), and FL-SDT (**D**). (**E**) Time courses showing the tumor diameters of the mice in (**A**–**D**) measured via MRI. (**F**) K–M survival curves of untreated mice and mice treated with FUS only, NaFL only, or NaFL-SDT (*n* = 6/group). Overall survival statistical analysis was performed using the Mantel–Cox log-rank test.

**Figure 4 cancers-16-00792-f004:**
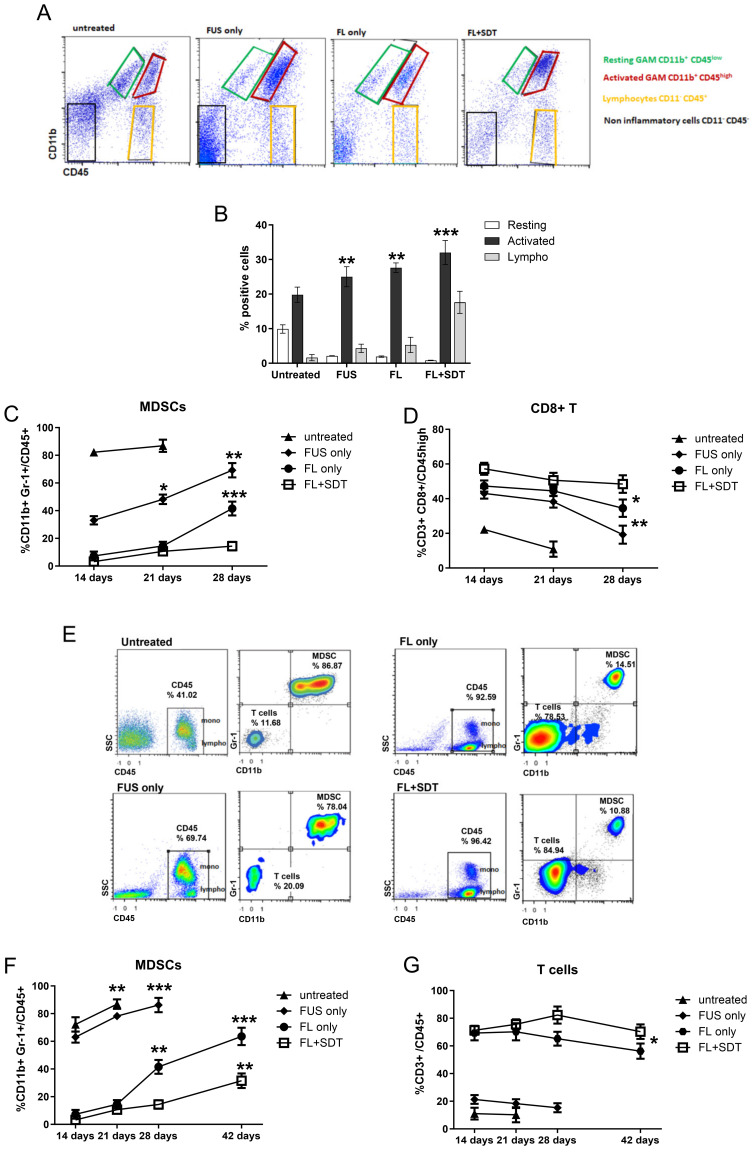
(**A**) Representative dot plots showing the immune cell infiltration assessed using CD11b and CD45 as markers. (**B**) Bar graphs showing the percentages of the different subsets of infiltrating immune cells, including resting and activated tumor-associated macrophages and lymphocytes. (**C**) Percentage of infiltrating MDSCs assessed by flow cytometry as CD11b+Gr1+ on the CD45+ gate in untreated and treated glioma tissues at 14, 21, and 28 days after tumor implantation. (**D**) Percentage of infiltrating CD8+ T cells assessed by flow cytometry as CD3+ CD8+ cells in the CD45high gate in untreated and treated glioma tissues at 14, 21, and 28 days after tumor implantation. (**E**) Representative dot plot showing the lymphocyte and monocyte counts on SSCs and the percentage of peripheral MDSCs (CD11b+ Gr+) in untreated and treated mice. (**F**) Time course of peripheral MDSCs in all the groups at the three time points evaluated for intratumoral quantification and later time points for FL−only and FL−SDT. (**F**,**G**) Time course of peripheral T cells in all the groups at the three time points evaluated for intratumoral quantification and later time points for FL−only and FL−SDT. (* *p* < 0.01, ** *p* < 0.005, *** *p* < 0.0001). The data from three experiments are presented as the mean ± SD.

## Data Availability

The data presented in this study are available on request from the corresponding author.

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
