# Peer review of "The Immunomodulatory Effects of Fluorescein-Mediated Sonodynamic Treatment Lead to Systemic and Intratumoral Depletion of Myeloid-Derived Suppressor Cells in a Preclinical Malignant Glioma Model"

_cancers, 2024, doi:10.3390/cancers16040792_

Round 1

Reviewer 1 Report

Comments and Suggestions for Authors

The authors present an in vitro and in vivo murine study of sonodynamic therapy for glioma treatment. The content is timely given clinical trials running in this area, and the investigations into immunomodulatory effects should stimulate further research. The main concern is the lack of ultrasound methodological description and thereby weakening the value of the data presentation. It should be straightforward for the authors to remedy this and present a suitably impactful paper. Specific comments follow.

Abstract: Add quantitative statements to support the findings of the study, instead of generalizations like 'tumor control and improved survival'.

L153 - Much more detail is need on the FUS instrumentation used for the in vitro and in vivo work. For example: 

i) How were the cells exposed (boundary conditions, in situ calibration?). 

ii) How were the mice positioned during exposure and was anything done to suppress unwanted reflections? 

iii) What pressures were used? 

iv) What was the beam pattern of the FUS source? 

v) Were microbubbles used in this study, and if not, what aspect of the ultrasound field activates the sensitizer, and how was this confirmed?

L156 - clarify specifically what was being sought/quantified with the MRI: evidence of BBB opening, lack of direct ablation from the FUS procedure?

Fig 3E, show the pre-treatment tumour sizes. What were the day 8 sizes for all mice?

L394: In what specific ways was safety assessed in this study? Please replace 'impressive' with a more appropriate / objective term.

Author Response

Plase see the attachment

Reviewer 2 Report

Comments and Suggestions for Authors

I found this manuscript very interesting due to development of the combined strategy and easy to repeat unsophisticated experiments. Such examination and future clinical trial phase could improve the quality of cancer patients treatment.

Why did the Authors choose only fluorescein and not the mentioned 5-ALA? Both compounds accumulate selectively in brain tumors and show good safety. If they had done comparative research, it might have turned out that brain tumors other than HGG exhibit uptake of 5-ALA stimulated by sonication. What is the mechanism of this limitation on accumulation in selected lesions?

Line 152: “mice were sonicated 20 minutes after” - Complete organisms? In what medium? Can you show the setup in the Supplementary Materials?

Figures:

Nowadays, much attention is paid to visualization in the presentation of results. Your results are good, but their presentation is too weak. All flow cytometers dot plots are difficult to analyze. I understand that this may be limited by the format of the data being acquired from the cytometer. Can the Editor help somehow?

Comments on the Quality of English Language

The manuscript is written in concise scientific language at an appropriate level enabling understanding of the Authors' intentions and experimental results. I have no objections to the language aspect of the work.

Round 2

Reviewer 1 Report

Comments and Suggestions for Authors

The authors made several useful updates to the manuscript. There are a two remaining questions to resolve:

L174: State the ultrasound pressure used for the in vitro experiment. Exposure frequency and time by themselves are not adequate descriptors of the treatment.

L368: Somewhere in the discussion, it is worth clarifying that this work is done without bubbles or other exogenous cavitation nuclei. This simplification is a potentially useful feature and likely reduces risks associated with cavitation that is employed in many SDT studies.

Comments on the Quality of English Language

Just minor editing required

Author Response

We thank the reviewer for the helpful suggestions. We have now addressed the comments, modifying the manuscript accordingly.

The revisions are highlighted in yellow in the revised manuscript.

Rev. State the ultrasound pressure used for the in vitro experiment. Exposure frequency and time by themselves are not adequate descriptors of the treatment.

Response. The Peak Positive Pressure employed was 100 kPa. This information is now present in Methods (L154, paraghaph 2.4).

Rev. Somewhere in the discussion, it is worth clarifying that this work is done without bubbles or other exogenous cavitation nuclei. This simplification is a potentially useful feature and likely reduces risks associated with cavitation that is employed in many SDT studies.

Response. We now added the clarification in the discussion (L404 – L409):

‘In our study, SDT treatment was performed without microbubbles considering that the energy provided by low-intensity focused ultrasound can activate fluorescein. Chemically, the main mechanism of action is that the energy released leads to production of reactive oxygen species (ROS), starting a peroxidation chain that leads to necrosis and apoptosis of tumor cells. However, SDT mechanisms of action are still under investigation.’